# Comparative Transcriptomics Reveal the Mechanisms Underlying the Glucosinolate Metabolic Response in Leaf *Brassica juncea* L. under Cold Stress

Bing Tang, Bao-Hui Zhang, Chuan-Yuan Mo, Wen-Yuan Fu, Wei Yang, Qing-Qing Wang, Ning Ao, Fei Qu, Guo-Fei Tan [ID], Lian Tao and Ying Deng *

Institute of Horticulture, Guizhou Academy of Agricultural Sciences/Horticultural Engineering Technology Research Center of Guizhou, Guiyang 550006, China; tangbin202208@163.com (B.T.); 18786102250@163.com (B.-H.Z.); mochuanyuan@163.com (C.-Y.M.); fuwenyuan@webmail.hzau.edu.cn (W.-Y.F.); yangwei139@sina.cn (W.Y.); wangqing92713431@sina.com (Q.-Q.W.); aoning2013@163.com (N.A.); 18798006209@163.com (F.Q.); tagfei@foxmail.com (G.-F.T.)
* Correspondence: dengyingstrong@sina.com

**Abstract:** Glucosinolates (GSLs) are not only a unique flavor substance from leaf *B. juncea* but also a major secondary metabolite produced in response to abiotic stresses. Cold stress is one of the most common abiotic stresses in leaf *B. juncea*; however, the metabolic response pattern of GSLs in leaf *B. juncea* under cold stress has not yet been reported. In the present study, we analyzed the GSLs content of leaf *B. juncea* under cold stress and found that it increased and subsequently decreased. According to RNA-seq data, genes related to the synthesis of aliphatic GSLs were significantly upregulated following 24 h of cold stress; genes related to the synthesis of indole GSLs were significantly upregulated following 48 h of cold stress; and *BjBGLU25* and *BjBGLU27* were significantly upregulated. Further analysis of the correlation between transcription factors and GSLs content revealed that MYB, ERF, IQD, and bHLH may be involved in regulating the GSLs response pattern in leaf *B. juncea* under cold stress. In particular, an unreported transcription factor, BjMYBS3 (BjuVA05G33250), was found to play a possible role in the synthesis of aliphatic GSLs. And the external application of GSLs increased the ability of leaf *B. juncea* to cope with cold stress.

**Keywords:** transcriptomics; *Brassica juncea* L.; glucosinolate; cold stress; transcription factor

## 1. Introduction

*Brassica juncea* L. is an annual or biennial herbaceous plant of the genus *Brassica* in the family Brassicaceae and is an important oil and vegetable crop in China [1]. *B. juncea* can be categorized on the basis of its edible organs into leaf *B. juncea*, stem *B. juncea*, and root *B. juncea*. [2,3]. Leaf *B. juncea* prefers to be cool, with a growth temperature of 15~22 °C, and optimum growth temperature is hindered at temperatures below 12 °C. Leaf *B. juncea* is mainly grown in the Yangtze River valley and the southwest of China, where it thrives in the cooler climate and higher altitude [2,4,5]; however, few studies exist on the physiological and biochemical characteristics of leaf *B. juncea* under cold stress, which is an abiotic stress factor that seriously affects the growth and development of plants [6,7]. In response to low-temperature stress, plants undergo dramatic changes in the expression of genes that regulate the metabolome, exhibiting different physiological and biochemical responses that ultimately lead to changes in membrane stability and the levels of osmoregulatory substances (proline), soluble sugars, amino acids, and secondary metabolites [7].

Glucosinolates (GSLs) are sulfur-containing secondary metabolites that play an important role in abiotic stress, plants resistance to insects and pathogens, and also display anti-cancer properties [8–12]. In plants, GSLs side chains are commonly derived from aliphatic and indole amino acids [11,12], such as methionine and tryptophan, respectively [8,9]. The biosynthetic pathways of methionine-derived aliphatic GSLs and

tryptophan-derived indole GSLs have been well-studied in cruciferous plants over the past few decades [9,10,13–15]. The aliphatic GSLs biosynthesis pathway has been shown to consist of three stages: side-chain extension, core structure formation, and side-chain modification [16]. *Branched-chain amino acid aminotransferase 4* (*BCAT4*), *BCAT3*, and *methylthioalkylmalate synthase* (*MAMs*) have been demonstrated to affect the diversity of aliphatic GSLs side-chain lengths [10,17,18]. Additionally, *CYP81Fs* catalyze the hydroxylation of indole GSLs [8,10,19]. Moreover, transcription factors also play an important role in the synthesis of GSLs. For instance, the overexpression of MYB28, -29, or -76 leads to the increased accumulation of aliphatic GSLs and the inhibition of the indole GSLs biosynthesis pathway [20–22]. MYC2, -3, and -4 basic helix–loop–helix (bHLH) transcription factors, produced in response to plant stress, and AP2 (ERF) and IQD, involved in ethylene signaling, play a role in the synthesis of GSLs [23–25]. GSLs are the main secondary metabolites produced in cruciferous plants in response to stress, and their biosynthesis is regulated by environmental factors. The involvement of GSLs in the response to abiotic and biotic stresses, such as plant diseases and insect pests, drought, salt damage, high temperature, circadian rhythm, and nutrient deprivation, has been studied in detail; however, the response processes under low-temperature stress remain elusive [12,26–28].

To elucidate the response mechanism of GSLs in leaf *B. juncea* under cold stress, we analyzed the transcriptome at 0 h (CK), 24 h (LT24), and 48 h (LT48). Additionally, we used the competition method to systematically detect the GSLs content in these samples. We identified the genes involved in GSLs synthesis and metabolism and correlated this information with the expression levels of related transcription factors to explore the response mechanism of GSLs under low-temperature stress. These data lay a solid foundation for the selection, development, and utilization of leaf *B. juncea* resources.

## 2. Materials and Methods

### 2.1. Plant Material and Cold Stress Treatment

The leaf *B. juncea* "Qianqing 6" was used in the present study, and cultivated at Institute of Horticulture, Guizhou Academy of Agricultural Sciences (Guiyang, China) under natural light and photoperiod. For cold stress treatment, we designed the following experiments. Three-month-old leaf *B. juncea* was transferred to a constant-temperature incubator at 22 °C for one week under optimal conditions. Subsequently, the temperature was adjusted to 4 °C and the same parts of the leaves were taken as samples at three time points: 0 h, 24 h, and 48 h. There were at least three biological replicates of the samples at each time point. Samples were used for GSLs content analysis, and the remaining were immediately frozen in liquid nitrogen and stored at −80 °C for further transcriptomics and PCR analyses.

For the GSLs topical application treatment, we designed the following experiment. Under optimal conditions, 3-month-old Monarch leaves were transferred to a constant temperature incubator at 22 °C for 7 d. Subsequently, the temperature was adjusted to 4 °C, and the control group was sprayed with water and the experimental group was sprayed with a GSLs solution at a concentration of 100 ng·L$^{-1}$. Both were sprayed until the liquid naturally dripped from the leaves, and leaves from the same sites were taken as samples after 14 d, samples for physiological indicators, and PCR analysis.

### 2.2. GSLs Enzyme Immunoassay

After the leaves were collected, they were repeatedly frozen and thawed at −20 °C three times and then filtered through glass fiber, extracted with butanol/methanol/water (5:25:70, V:V:V) as the sample to be tested; standard, blank (no enzyme reagent samples), and sample wells were prepared on the enzyme plate. We added 50 μL of GSLs standard solution to the standard well, 40 μL of sample dilution to the sample well, then added 10 μL of the sample solution to be tested to the sample well (the final dilution is 5×). Subsequently, 50 μL enzyme standard reagents were added to each well, except for the blank wells. The plate was sealed with film and incubated at 37 °C for 1 h; meanwhile, the

30-fold concentrated wash solution was diluted with distilled water. The sealing membrane was then carefully removed, the liquid discarded, and the plate shaken dry. The wells were washed with wash solution, allowed to incubate for 30 s, and the solution was then discarded; this was repeated 5 times and the plate was then patted dry. Next, 50 μL chromogenic agent A and 50 μL chromogenic agent B were added to each well and the plate was gently shaken to mix. The reaction was allowed to develop for 10 min at 37 °C in the dark. A 50 μL aliquot of termination solution was added to each well to terminate the reaction (the blue color turned yellow). Within 15 min of termination, the absorbance (OD) of each well was measured at 450 nm and normalized to the blank wells. Using the concentrations of the standard as the horizontal coordinates and the OD values as the vertical coordinates, the standard curve was plotted, and the corresponding concentrations were found and then multiplied by the dilution factors.

### 2.3. Physiological Analyses of Cold-Treated Leaves

Four cold response indicators, soluble sugars, soluble proteins, malondialdehyde, and the proline contents of leaf *B. juncea* leaves, were determined. Soluble sugar levels were determined using the anthranilate method [29]. The malondialdehyde content was determined via the thiobarbituric acid reaction method [30]. The proline concentration was determined by the sulfosalicylic acid–acidic ninhydrin method [31]. Soluble protein concentrations were determined using the bis(urea) method. All the above determinations were performed using Solarbio commercial kits (Solarbio Co., Ltd., Beijing, China).

### 2.4. RNA Sequencing (RNA-Seq) and Data Analysis

Leaves (three biological replicates) from the control and cold stress groups were sent to Guizhou Shenglangsai Biotechnology Co., Ltd. (Guiyang, China) for RNA-seq. Total RNA was isolated using the RNAiso Plus kit (TaKaRa, Dalian, China) and analyzed using the RNA Nano 6000 assay kit (Agilent Technologies, Santa Clara, CA, USA) on a NanoDrop™ 2000 (Thermo Scientific, Waltham, MA, USA) to assess the RNA concentration and integrity. Approximately 1 μg RNA was used to construct the cDNA libraries, the quality of which was assessed on an Agilent Bioanalyzer 2100 system. The prepared libraries were sequenced on the Illumina HiSeq platform to generate raw reads of paired ends. After data processing, reads were filtered to remove adapters, reads containing poly(N), and low-quality reads. High-quality reads were mapped to the leaf *B. juncea* (T84-66) reference genome [32] using HISAT2 v. 2.2.0. (https://ccb.jhu.edu/software/hisat2/index.shtml (accessed on 6 October 2022)). Gene expression levels were calculated using the FPKM (fragments per kilobase of transcript per million fragments mapped) method. Differentially expressed genes (DEGs) were identified using the DESeq R package V 1.24.0 based on $|\log2 \text{ (fold change)}| \geq 1$ and false discovery rate (FDR) < 0.01. GO (Gene Ontology) enrichment of DEGs and KEGG (Kyoto Encyclopedia of Genes and Genomes) pathway enrichment analysis was performed using the cluster Profiler 4.0 software [33].

### 2.5. Quantitative Real-Time Reverse Transcription PCR (qPCR)

Total RNA for qPCR was extracted as described above. First-strand cDNA was synthesized using the PrimeScript First-Strand cDNA Synthesis Kit (TaKaRa, Dalian, China) according to the manufacturer's instructions. The 10 μL reaction solution for qPCR analysis contained 100 ng cDNA, 0.25 μM forward and reverse primers, and 5 μL SYBR Green Master Mix (TaKaRa, Dalian, China). *BjACTIN* was used as an internal reference. Gene-specific primers (Table S8) were designed using the IDT tool (https://sg.idtdna.com/scitools/Applications/RealTimePCR/Default.aspx (accessed on 6 October 2022)). The $2^{-\Delta\Delta CT}$ method was used to calculate the relative expression levels. Three technical replicates were performed for each sample.

### 2.6. Statistical Analysis

Statistical data are expressed as the mean ± standard error. A Student's *t*-test and one-way analysis of variance (ANOVA) were used for statistical analysis (statsmodelsb package), and differences were considered significant at * $p < 0.05$, ** $p < 0.01$, and *** $p < 0.001$. Pearson's correlation coefficient was calculated using the statsmodels package. Heatmaps were plotted using TBtools [34].

## 3. Results

### 3.1. Difference in GSLs Content in Leaf B. Juncea under Cold Stress

Leaf *B. juncea* was subjected to cold stress treatment at 4 °C and samples were taken at 0 h (control group, denoted as CK), 24 h (denoted as LT24), and 48 h (denoted as LT48) to measure the total GSLs content (Figure 1A). Comparison of the total GSLs content in the leaves of leaf *B. juncea* among the three time points demonstrates that the GSLs content in LT24 (66.38 ng·kg$^{-1}$) and LT48 (63.52 n·kg$^{-1}$) leaves was significantly higher than that in CK leaves (59.89 ng·kg$^{-1}$), and that the GSLs content in LT24 leaves was significantly higher than that in LT48 leaves (Figure 1B). These results indicate that under cold stress, the GSLs content of leaf *B. juncea* leaves initially increased; however, as the time under cold stress increased, the GSLs content in the leaves began to decrease.

A
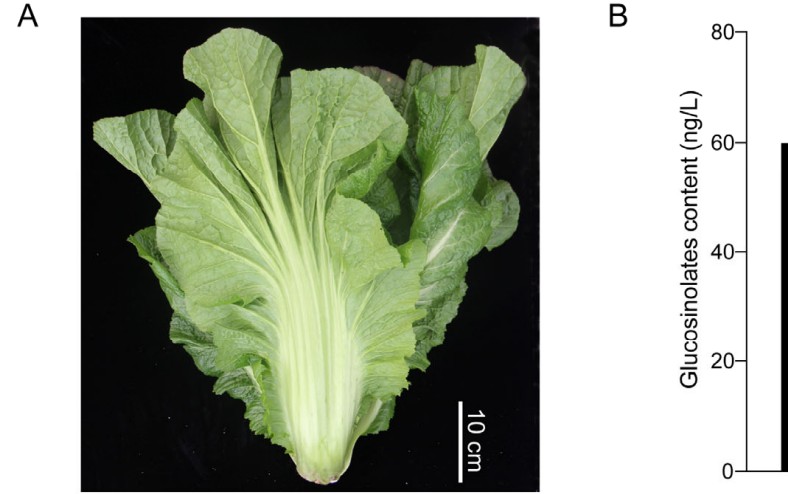

B
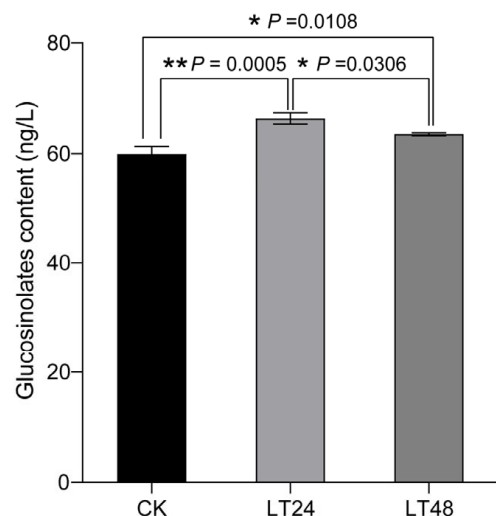

**Figure 1.** GSLs content of leaf *B. juncea* under cold stress at 0 h, 24 h, and 48 h. (**A**) Leaf *B. juncea* "Qianqing 6". (**B**) GSLs content (* $p < 0.05$; ** $p < 0.01$).

### 3.2. Sequencing and Assembly of RNA-Seq Datasets

Transcriptome sequencing data were used to elucidate the molecular mechanism underlying changes in the GSLs content of leaf *B. juncea* leaves under cold stress. Illumina RNA-seq analyzed biological replicates of leaf *B. juncea* leaves under cold stress at three time points, and each sample produced 38.42~50.64 million clean reads after quality control, with the Q30 exceeding 92.90%. In addition, 35.21~40.80 million clean reads were mapped to the mustard reference genome (90.73~92.52% mapping rate) (Table S1).

We clustered samples using principal component analysis (PCA) based on the expression of all genes in the sample. PCA1 shows that the three biological replicates of samples CK, LT24, and LT48 were clustered together individually, and PCA2 shows little in-sample variation among the three samples, in addition to high sample consistency (Figure S1), which indicates that the gene expression profiles of the samples were highly consistent.

### 3.3. Identification and Enrichment Analysis of DEGs

To identify differentially expressed genes (DEGs) in response to cold stress in the leaves of leaf *B. juncea*, the genes expression profile at 24 h and 48 h under cold stress was compared with that at 0 h: CK vs. LT24 and CK vs. LT48 (Figure 2A). A total of

10,448 DEGs were identified in leaf *B. juncea* leaves following 24 h of cold stress compared with the control (CK vs. LT24), whereas 16,133 DEGs were identified following 48 h of cold stress (CK vs. LT48, Figure 2B). A total of 8331 DEGs were identified from these two comparisons, of which 4585 were jointly upregulated, 3693 were jointly downregulated, and 53 were regulated in different directions. Moreover, 4175 upregulated DEGs and 3627 downregulated DEGs occurred only in CK vs. LT48, and 1130 upregulated DEGs and 1054 downregulated DEGs were specific to CK vs. LT24. We created an UpSet plot to illustrate the number of DEGs in the two comparisons (Figure 2C). To independently assess the reliability of the RNA-seq data, the expression patterns of 20 randomly selected genes were analyzed using RT-qPCR, which were highly correlated ($R^2$ = 0.7886) with the RNA-seq results, indicating data reliability (Figure S2). These results imply that a greater number of genes begin to respond to cold stress as the duration increases, and that DEGs in both comparisons may be associated with changes in the GSLs content in leaf *B. juncea* leaves during cold stress.

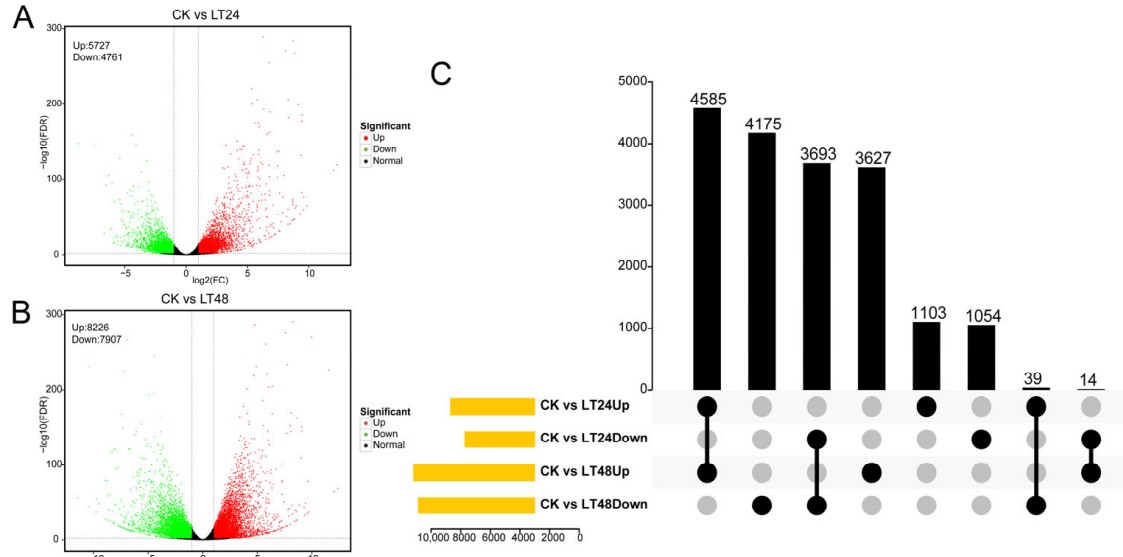

**Figure 2.** Volcano plots of genes expression and UpSet plots of DEGs in the CK vs. LT24 and CK vs. LT48 comparisons. (**A**) Volcano plots of CK vs. LT24 genes. (**B**) Volcano plots of CK vs. LT48 genes. Red, green, and black dots indicate upregulation, downregulation, or no difference in gene expression, respectively. (**C**) UpSet plots of the DEGs in the CK vs. LT24 and CK vs. LT48 comparisons.

To further understand the potential mechanisms underlying changes in the GSLs content in the leaves of leaf *B. juncea*, under cold stress, we performed GO (Gene Ontology) and KEGG (Kyoto Encyclopedia of Genes and Genomes) enrichment analysis of the DEGs for both CK vs. LT24 and CK vs. LT48 comparisons (Figure 3). The 50 most enriched GO terms from biological processes, cellular components, and molecular functions were determined, of which 49 were identical. GO terms were more enriched to DEGs in the CK vs. LT48 comparison. The most abundant GO terms for biological processes of common DEGs were cellular processes, metabolic processes, single-organism processes, response to stimuli, and biological regulation. The most abundant GO terms for cellular components of common DEGs were cells, cell parts, organelles, membranes, membrane parts, organelle parts, and macromolecular complexes. The most abundant GO terms for molecular functions of common DEGs were binding, catalytic activity, transporter activity, structural molecule activity, nucleic-acid-binding transcription factor activity, and signal transducer activity (Figure 3A,B). Among them, response to stimuli, biological regulation, nucleic-acid-binding transcription factor activity, and signal transducer activity are likely associated with the synthesis of GSLs. The standard pathway enrichment analysis based on KEGG differed from GO enrichment in that only 11 of the 20 most-enriched KEGG pathways were common to both comparisons. Ribosomes associated with protein synthesis and photosynthesis

(ko03010), ribosome biogenesis in eukaryotes (ko03008), photosynthesis–antenna proteins (ko00196), photosynthesis (ko00195), and RNA transport (ko03013) pathways were the most enriched with the highest number of DEGs in both comparisons (Figure 3C,D). In addition, the GSLs biosynthesis (ko00966) pathway was enriched in both comparisons; however, in the CK vs. LT24 comparison, the GSLs biosynthesis pathway was enriched to 11 DEGs with a higher enrichment factor of 2.46 (Figure 3C), whereas in the CK vs. LT48 comparison, the GSLs biosynthesis pathway was only enriched to 5 DEGs with an enrichment factor of 0.58 (Table S1). This may be the reason for the change in the GSLs content.

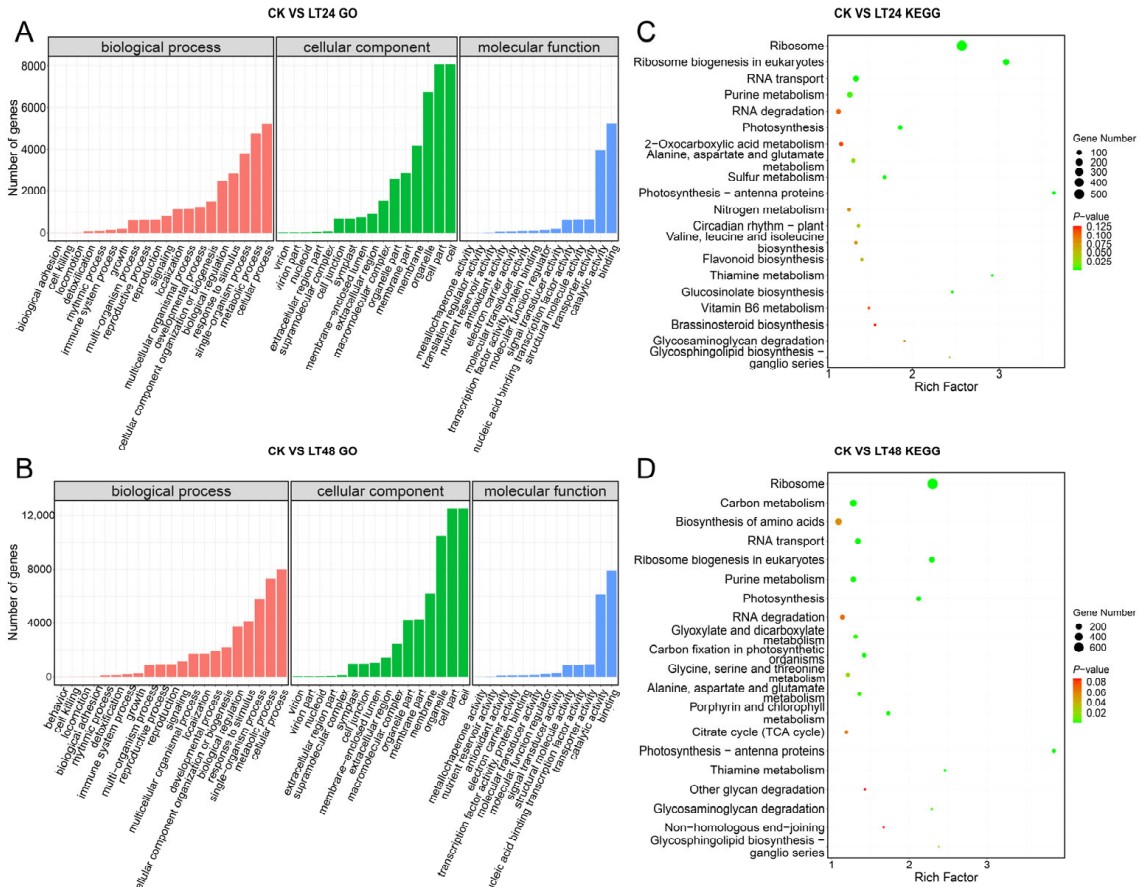

**Figure 3.** GO and KEGG enrichment analysis of DEGs in the CK vs. LT24 and CK vs. LT48 comparisons. (**A**,**B**) The 50 most significantly enriched GO terms for DEGs in the CK vs. LT24 and CK vs. LT48 comparisons. (**C**,**D**) The 20 most significantly enriched KEGG pathways for DEGs in the CK vs. LT24 and CK vs. LT48 comparisons. The color and size of the dots indicate the Q value and the number of genes, respectively.

### 3.4. Expression Patterns of Structural DEGs Related to GSLs Biosynthesis

To further elucidate the mechanism underlying changes in GSLs content in *B. juncea* under cold stress, we remapped the GSLs synthesis pathway in *B. juncea* (Figure 4A) based on the *Arabidopsis* GSLs synthesis pathway, KEGG database, and related literature [12,16,35,36]. In total, we identified 29 major genes in the GSLs synthesis pathway in leaf *B. juncea* (Figure 4, Tables S2–S4). In the CK vs. LT24 comparison, 11 DEGs were present among the 29 synthesized genes, which encoded a branched-chain amino acid aminotransferase (BCAT, *BAT4*: BjuVB01G34920), a methylthioalkylmalate synthase (MAM, *MAM1*: BjuVA03G45980), three cytochrome P450s (CYP, *CYP79B3*: BjuVB01G18500, *CYP79F1*: BjuVA06G12040, *CYP79F2*: BjuVB06G35270), two glutathione S-transferases (GST, *GST9*: BjuVB01G11760, *GST20*: BjuVA07G27800), a C-S lyase (SUR, *SUR1*: BjuVB03G59170), two sulfotransferases (SOT, *SOT17*: BjuVA06G14130, *SOT18*: BjuVB03G46710), and an

alkenyl-hydroxalkyl-producing protein (AOP, *AOP3*: BjuVB05G45660). In total, 9 of these 11 DEGs were upregulated and all were involved in the synthetic pathway of aliphatic GSLs. *CYP79B3* (BjuVB01G18500) and *GST9* (BjuVB01G11760) were downregulated and upregulated, respectively, which play a role in the synthetic pathway of indole GSLs (Figure 4A, Table S2). In the CK vs. LT48 comparison, 10 DEGs were present out of 29 synthesized genes. Unlike the CK vs. LT24 comparison, the majority of these 10 DEGs were upregulated and associated with the synthesis of indole GSLs, namely *CYP79B3* (BjuVB01G18500), *GSTF9* (glutathione S transferase F9, BjuVB01G11760), *GSTF10* (glutathione S-transferase F10, BjuVA03G16960), *SUR1* (C-S lyase 1, BjuVB03G59170), CYP81F1 (cytochrome P450 81F, BjuVB05G01380), *CYP81F3* (cytochrome P450 81F3, BjuVA01G01560), and *IGMT1* (indole GSL O-methyltransferase 1, BjuVB04G32680). Among the DEGs associated with the synthesis of aliphatic GSLs, *CYP81A1* (cytochrome P450 81A1) and *GSTF11* (glutathione S-transferase F11) expression was downregulated and *SOT17* (sulfotransferase 5c) expression was upregulated (Figure 4A, Table S2). Interestingly, most of the differential gene expression in the synthesis pathway of aliphatic GSLs showed a trend to increase at LT24 and downregulate at LT48, whereas most of the differential gene expression in the synthesis pathway of indole GSLs showed a trend to be consistently up-regulated.

In addition, we focused on GSLs-degradation-related genes, *BGLU* (β-glucosidase), known as black mustard enzymes. We identified a total of 46 typical and 49 atypical black mustard enzymes in the *B. juncea* genome based on *Arabidopsis thaliana*, which possesses 6 typical and 16 atypical black mustard enzymes [37,38]. In the CK vs. LT24 comparison, only six DEGs were *BGLUs*: two typical *BGLUs* were downregulated and three were upregulated, all belonging to *BGLU34–36*, and one atypical *BGLU*, *BGLU27*, was upregulated (Figure 4B, Table S5). In the CK vs. LT48 comparison, 16 DEGs were *BGLUs*: 12 typical *BGLUs* belonging to *BGLU34–36* and 3 atypical *BGLUs*, 2 *BGLUs25s* and 1 *BGLU27*, were upregulated, and 1 atypical *BGLU*, *BGLU33*, was downregulated in (Figure 4B, Table S6). Upregulated *BGLUs* may be involved in the degradation of GSLs.

### 3.5. Candidate Transcription Factors Involved in GSLs Biosynthesis

Four families of transcription factors (TFs), MYB, ERF, IQD, and bHLH, have been reported to be involved in the synthesis and degradation of GSLs [20,25,39–41]. Accordingly, we performed Pearson correlation coefficient analysis of the expression of DEGs that were members of these families (Figure 5, Tables S6 and S7) and identified homologs of MYB28, MYB51, IQD1, and ERF107 that regulate the synthesis of GSLs. Among them, MYB28 (BjuVA02G46870) was negatively associated with two atypical BGLUs and GSTF9 and significantly positively associated with two typical BGLUs. Three MYB51 homolog genes, BjuVB04G30350, BjuVA08G30790, and BjuVB03G26160, were identified, all of which were significantly positively correlated with the genes involved in indole GSLs synthesis, especially BjuVB04G30350, which was significantly positively correlated with the three transcription factors involved in indole GSLs synthesis, CYP79B3, CYP81F3, and CYP81F1 (Figure 5 and Table S6). There were six IQD1 homolog genes: BjuVA01G43990, BjuVA06G26910 BjuVA08G10210, BjuVB08G33460, BjuVA10G33380, and BjuVA10G00680. Notably, BjuVA01G43990 was significantly positively associated with five genes involved in indole GSLs synthesis: *CYP79B3, CYP81F1, IGMT1, GSTF10,* and *CYP81F3*. In addition, two IQD1 homologs, BjuVA10G00680 and BjuVB08G33460, were significantly negatively correlated with total GSLs. We also found that the TFs MYBS3 (BjuVA05G33250) was significantly positively correlated with four aliphatic GSLs synthesis genes, *MAM1, AOP3, GSTU20,* and *CYP79F1*, in addition to the total GSLs content. There were also seven TFs, MYB (BjuVA03G36050, BjuVB04G18620, BjuVB03G47230), ERF (BjuVA08G21510, BjuVA08G20650), and bHLH (BjuVA03G01500, BjuVB06G55140), that were significantly positively correlated with indole GSLs synthesis genes, and the MYB TF BjuVA06G03530 was significantly negatively correlated with the genes involved in the indole family GSLs synthesis (Figure 5 and Table S6).

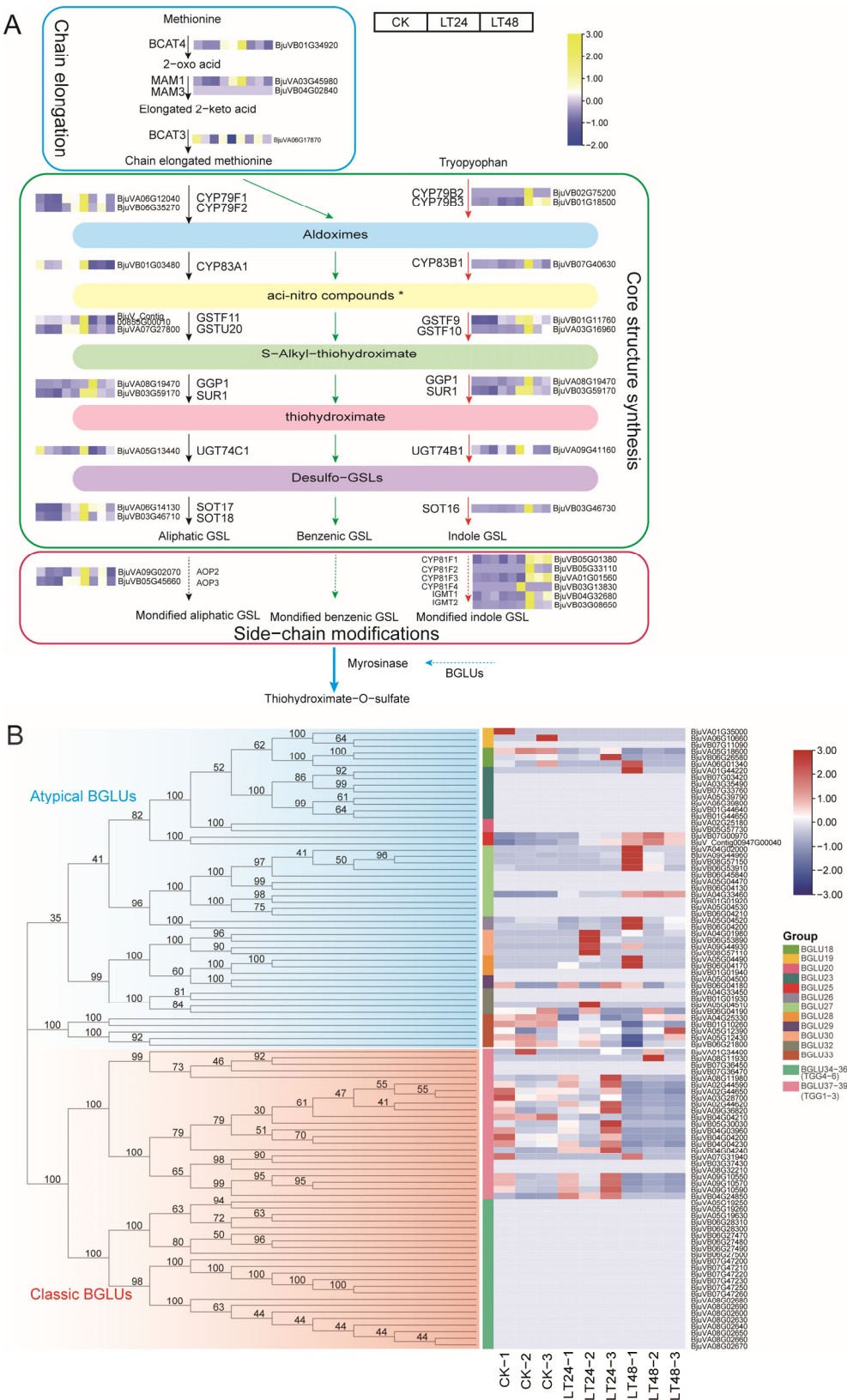

**Figure 4.** The GSLs synthesis pathway and related genes expression. (**A**) Construction of the GSLs synthesis pathway and synthetic gene expression in *B. juncea*. (**B**) Evolutionary tree produced using the neighbor-joining method and gene expression of the GSLs-degradation-related genes, BGLUs.

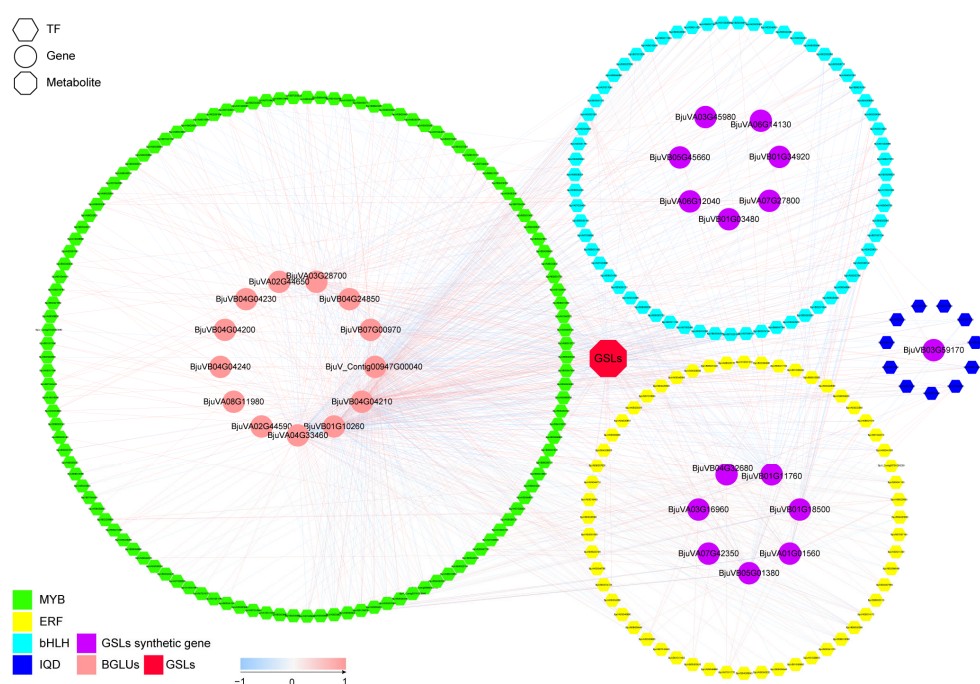

**Figure 5.** Correlation of MYB, ERF, bHLH, and IQD gene family TFs with genes of *B. juncea* related to GSLs synthesis and content.

### 3.6. Effect of Topical Application of GSLs on Cold Resistance in Leaf B. juncea

The leaves color of leaf *B. juncea* became lighter after 14 d of cold stress (Figure 6A). Regarding the cold-stress-related indicators, the malondialdehyde (MDA), free proline, and soluble sugar contents in the leaves of leaf *B. juncea* with external application of GSLs were significantly lower than those of the control group, and the soluble protein content was significantly higher than that of the control group (Figure 6B–E). We also examined the expression of GSLs-synthesis-related genes with significantly elevated expression in the transcriptome. The expression of *GSTF9* was significantly lower, and the expressions of *CYP79B3*, *CYP79F1*, *CYP79F2*, *GSTF10*, *SUR1*, *SOT17*, *SOT18*, and *MYBS3* were significantly higher in the leaves of leaf *B. juncea* with the external application of GSLs than in the control group, indicating that GSLs were elevated in the leaves of leaf *B. juncea* during cold stress after the external application of GSLs.

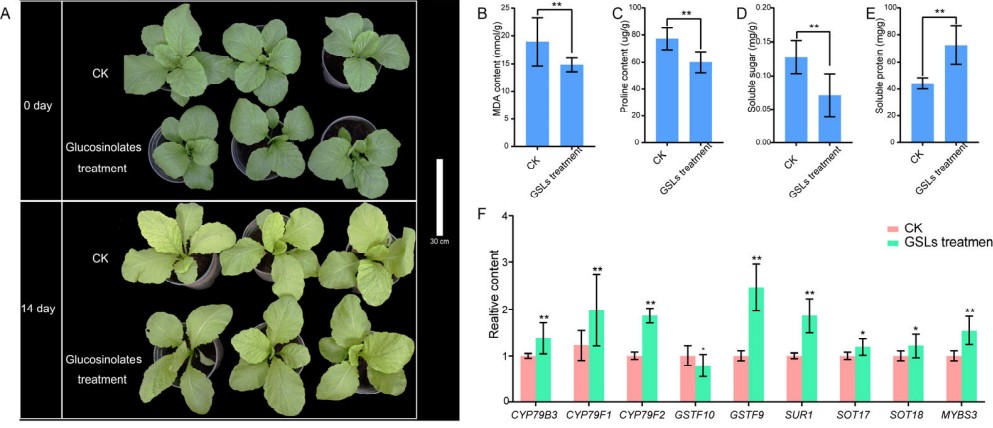

**Figure 6.** Effect of GSLs treatment on cold resistance in leaf *B. juncea*. (**A**) Phenotypes of GSLs-treated *B. juncea* and control after 14 d of cold stress; (**B–E**) propylene glycol, free proline, soluble sugar, and soluble protein contents of GSLs-treated *B. juncea* and control after 14 d of cold stress; (**F**) Expression of GSLs-synthesis-related genes (*CYP79B3*, *CYP79F1*, *CYP79F2*, *GSTF9*, *GSTF10*, *SUR1*, *SOT17*, *SOT18*, and *MYBS3*) in GSLs-treated *B. juncea* and control after 14 d of cold stress (* $p < 0.05$; ** $p < 0.01$).

## 4. Discussion

GSLs are special pungent flavor substances in cruciferous plants, such as *B. juncea* [42], but recent studies have shown that GSLs also play an important role in the response to biotic stresses, such as insect feeding and pathogen infestation, as well as abiotic stresses such as water, light, and temperature [43]. In the present study, we focused on changes in the GSLs content of leaf *B. juncea* leaves under cold stress, which initially increased with time but subsequently began to decrease. However, even though cold stress is the most common stress faced during leaf *B. juncea* growth and development [2], no relevant reports exist regarding the response pattern of GSLs in leaf *B. juncea* under cold stress. The results show that the response pattern of GSLs under cold stress is similar to that under high-temperature stress [44,45], whereby high temperatures promote the synthesis of GSLs, but extreme high-temperature stress decreases the GSLs content [46,47]. The GSLs content is one of the criteria for evaluating the quality of leaf *B. juncea*, and in actual production, the GSLs content and quality are increased by preserving leaf *B. juncea* at low temperatures for a short period of time.

A previous study revealed the molecular mechanisms underlying GSLs synthesis and response under abiotic stress, such as the transcriptional upregulation of MYB28/29 under drought stress, which further increases the transcript levels of the genes involved in GSLs synthesis, such as *MAM1*, *CYP79F1*, and *CYP83A1* [48]. Moreover, *BGLU30* has been shown to mediate the hydrolysis of GSLs under dark conditions [12]. Transcription factors, GSLs synthesis genes, and β-glucosidases (BGLUs) are all involved in GSLs synthesis and response under abiotic stresses. Here, we compared the transcriptome of leaf *B. juncea* under different temporal gradients of cold stress and show that global transcriptome changes occur. More unique DEGs were identified in the CK vs. LT48 comparison than in the CK vs. LT24 comparison, and GO enrichment results show that the 50 most-enriched GO terms related to the DEGs in both comparisons were essentially the same, suggesting that *B. juncea* responds to cold stress via the same mechanisms over time but requires a greater number of genetic responses to adapt to cold stress as time increases. The KEGG pathway enrichment results further reveal the effects of cold stress on *B. juncea*. Photosynthesis (ko03010, ko00195), ribosome biogenesis in eukaryotes (ko03008), photosynthesis–antenna proteins (ko00196), and RNA transport (ko03013) were the most enriched pathways in both comparisons, suggesting that plants adapt to cold stress by altering protein synthesis and photosynthesis, which is largely consistent with previous reports [49]. The KEGG enrichment results also tentatively explain the response pattern of GSLs under cold stress. The GSLs biosynthesis (ko00966) pathway was enriched to 11 DEGs in the CK vs. LT24 comparison, while only 5 DEGs were enriched in the CK vs. LT48 comparison. The initial increase and subsequent decrease in the number of DEGs under cold stress was directly responsible for the initial increase and subsequent decrease in GSLs content.

Three types of GSLs are produced in the order Brassicales: aliphatic, indole, and aromatic [16]. Aliphatic and indole GSLs are the major components of GSLs in leaf *B. juncea*, and the gene families involved in the synthesis of aromatic GSLs and the transcription factors regulating their synthesis remain unclear [13,15,16,32,50]; therefore, we focused on the former GSLs. In the CK vs. LT24 comparison, nine genes in the synthesis pathway of aliphatic GSLs were significantly upregulated: the chain-extension-related genes *MAM1* (BjuVA03G45980) and *BAT4* (BjuVB01G34920); the core-structural-synthesis-related genes *CYP79F1* (BjuVA06G12040), *CYP79F2* (BjuVB06G35270), *GST20* (BjuVA07G27800), *SUR1* (BjuVB03G59170), *SOT17* (BjuVA06G14130), and *SOT18* (BjuVB03G46710); and the side-chain-modification-related gene *AOP3* (BjuVB05G45660). However, only *CYP79B3* (BjuVB01G18500) and *GST9* (BjuVB01G11760) in the synthesis pathway of indole GSLs were significantly upregulated and downregulated, respectively. This indicates that the increased numbers of GSLs at 24 h following cold stress mainly consist of aliphatic GSLs, and the content of indole GSLs may be unchanged. Interestingly, however, despite the reduced content of GSLs at LT48 relative to LT24, not all genes in the GSLs synthesis pathway were downregulated. In the CK vs. LT48 comparison, seven indole group synthesis-related

genes were significantly upregulated, *CYP79B3* (BjuVB01G18500), *GSTF9* (BjuVB01G11760), *GSTF10* (BjuVA03G16960), *SUR1* (BjuVB03G59170), *CYP81F1* (BjuVB05G01380), *CYP81F3* (BjuVA01G01560), and *IGMT1* (BjuVB04G32680), while genes associated with aliphatic GSLs synthesis, such as *CYP81A1* (BjuVB01G03480) and *GSTF11* (BjuV_Contig00855G00010), were downregulated and *SOT17* (BjuVA06G14130) was upregulated. This indicates that although the total content of GSLs decreased, the indole GSLs fraction may have increased. It is known that there exists crosstalk between the biosynthesis of indole family GSLs and the synthesis of indole-3-acetic acid (IAA). IAA production is increased following the blockade of the synthesis of indole family GSLs [11,14,51,52]. Moreover, plant growth was inhibited under cold stress, and the IAA content was reduced in above-ground tissues [53], suggesting that plants may increase their indole GSLs content to antagonize IAA. To reveal the reasons for the reduced aliphatic GSLs content, we further analyzed GSLs-degradation-related genes. GSLs were hydrolyzed by a group of β-glucosidases (*BGLU*) called myrosinases [12]. The *BGLU* gene family was amplified in leaf *B. juncea*; however, most *BGLUs* were not expressed or expressed at low levels. Three atypical *BGLUs*, *BGLU25* (BjuVB07G00970, BjuV_Contig00947G00040), and *BGLU27* (BjuVA04G33460), were significantly upregulated at LT48. Although *BGLU27* (BjuVA04G33460) was significantly upregulated at both LT24 and LT48, the upregulation multiplier was higher at LT48; therefore, these three *BGLUs* may be involved in the degradation of GSLs, especially aliphatic GSLs.

The biosynthesis of GSLs was regulated by many different factors. The transcriptional regulation aspect was well known, for example, *MYB28*, *MYB76*, and *MYB29* transcription factors regulate the biosynthesis of aliphatic thioglycosides. Moreover, MYB51-overexpressing lines show the increased accumulation of indole-3-methyl glucosinolate, the ERF gene family activates the biosynthesis of indole family GSLs to enhance plant defense against Verticillium longum, and bHLH05 functions in the synthesis of indole family GSLs by interacting with MYB51 [20,24,25,40,41,54]. Our data demonstrate that transcription factor IQD1, MYB28, and MYB51 were significantly correlated with GSL-degradation genes, with IQD1 (BjuVA01G43990) and MYB51 (BjuVB04G30350) being significantly positively correlated with several indole family GSLs synthesis genes, and MYB28 being significantly negatively correlated with the GSL degradation genes *BGLU25* (BjuV_Contig00947G00040) and *BGLU27* (BjuVA04G33460). This suggests that IQD1, MYB28, and MYB51 may be involved in the regulation of GSLs biosynthesis in the leaves of leaf *B. juncea* under cold stress. We also found that the transcription factor MYBS3 (BjuVA05G33250) was positively correlated with multiple genes involved in aliphatic GSLs synthesis, and we suggest that MYBS3 (BjuVA05G33250) may be involved in the regulation of aliphatic GSLs biosynthesis in leaf *B. juncea* leaves under cold stress. In addition, several members of the MYB, ERF, and bHLH families showed significant positive correlations with indole group GSLs synthesis genes, and these transcription factors may be candidates for the regulation of indole group GSLs synthesis.

We also investigated the effect of the external application of GSLs on the cold resistance of leaf *B. juncea*. After 14 d of cold stress, although the color of GSLs-treated leaf *B. juncea* was similar to that of the control, malondialdehyde (MDA) was significantly lower than that of the control. Cell membrane stability is considered to be a reliable indicator of cellular damage caused by biotic and abiotic stresses [55]. In contrast, MDA content can be used to assess the degree of lipid peroxidation, thus reflecting the extent of oxidative damage to cells [56]. The lower MDA content of GSLs-treated leaf *B. juncea* leaves indicates that the external application of GSLs can improve the cold resistance of leaf *B. juncea*, and we speculate that the increase in GSLs content in cold stress may be a mechanism for leaf *B. juncea* to protect itself in response to cold stress. We examined the content of free proline, soluble sugars, and soluble proteins that affected the cold resistance performance, and only the content of soluble proteins was significantly higher in GSLs-treated leaf *B. juncea* leaves than in the control. Thus, we hypothesized that the external application of GSLs increased the increase in soluble protein content and improved its cold resistance. In addition, we also examined the genes related to the synthesis of GSLs, and the expression of most of

them was significantly higher in the leaves of leaf *B. juncea* after GSLs treatment than in the control, which indicates that GSLs were increased in leaf *B. juncea* leaves during cold stress after the external application of GSLs. This may be one of the reasons for the increased cold resistance in leaf *B. juncea*. However, the mechanisms by which the soluble protein and endogenous GSLs contents are increased by the external application of GSLs may have to be further investigated.

## 5. Conclusions

In summary, we analyzed the GSLs content and transcriptomics data of the leave of leaf *B. juncea* under cold stress, identified the key genes and transcription factors involved in the GSLs response to cold stress, and clarified the GSLs response patterns and molecular mechanisms. We conclude that the expression of genes related to the synthesis of aliphatic GSLs were upregulated during the early stages of cold stress, and the content of aliphatic GSLs increased, as did the total GSLs content. With increased time under cold stress, the expression of genes related to the synthesis of indole GSLs was upregulated and the content of indole GSLs increased; however, the expression of genes related to the synthesis of aliphatic GSLs was downregulated, the expression of BGLU25 and BGLU27 was upregulated, and the content of adipose GSLs decreased, leading to a decrease in the total GSLs content. Many transcription factors may be involved in the regulation of this process. In addition, we found that the external application of GSLs increased the ability of leaf *B. juncea* to cope with cold stress, which was associated with an increase in the content of endogenous GSLs and soluble proteins. These findings provide insight into the molecular basis of the GSLs response pattern under cold stress in leaf *B. juncea* and offer novel genetic information for screening new leaf *B. juncea* varieties with a high GSLs content.

**Supplementary Materials:** The following supporting information can be downloaded at: https://www.mdpi.com/article/10.3390/agronomy13071922/s1, Figure S1: PCA map of transcriptome samples; Figure S2: Validation of gene expression ratios between the RNA-Seq and qRT-PCR analyses; Table S1: Transcriptome quality control data; Table S2: Differential ploidy of related genes in the GSL synthesis pathway; Table S3: Differential gene and database annotation of CK vs. LT48; Table S4: Differential gene and database annotation of CK vs. LT48; Table S5: BGLU gene grouping and differential multiplicity; Table S6: Synthesis-related genes and transcription factor correlations in GSLs; Table S7: GSLs and transcription factor correlation; Table S8: qRT-PCR primer sequences.

**Author Contributions:** Conceptualization, Y.D. and B.T.; methodology, C.-Y.M.; validation, L.T., F.Q. and N.A.; formal analysis, B.T.; investigation, B.-H.Z., W.-Y.F., W.Y. and Q.-Q.W.; resources, Y.D.; data curation, B.-H.Z.; writing—original draft preparation, B.-H.Z.; writing—review and editing, B.T., Y.D. and G.-F.T.; visualization, B.-H.Z.; supervision, G.-F.T.; funding acquisition, Y.D. All authors have read and agreed to the published version of the manuscript.

**Funding:** This study was funded by the project of Guizhou Provincial Department of Science and Technology "Construction and Utilization of Horticultural Platform of Plant GenBank Creation " (No. Qiankehe Fuqi [2022] 005); Guiyang Vegetable Germplasm Resources Research Center Construction Project (Chikke Contract [2021] No. 5-1); Guizhou Science and Technology Support Program Project (Qian Kehe Support [2022] General 086), Guizhou Modern Agriculuture Research System (GZMARS)—Plateau Characteristic Vegetable Industry.

**Data Availability Statement:** All data generated or analyzed during this study are included in the manuscript and its additional files. The sequencing dataset used in the study is available in the Sequence Read Archive of the NCBI database under BioProject: PRJNA993419, which will be made public after publication.

**Conflicts of Interest:** The authors declare no conflict of interest.

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
