# Peer review of "Comparative Transcriptomics Reveal the Mechanisms Underlying the Glucosinolate Metabolic Response in Leaf Brassica juncea L. under Cold Stress"

_agronomy, doi:10.3390/agronomy13071922_

Round 1
Reviewer 1 Report
The manuscript entitled “Comparative Transcriptomics Reveal the Mechanisms Under-lying the Glucosinolate Metabolic Response in Leaf Brassica juncea L. Under Cold Stress” authored by tang et al. is interesting and meaningful. I have read the manuscript and suggest some minor revisions before to accept the paper for publication in Agronomy.
1. How come the analysis in the DEG differential analysis in lines 186-201 did not analyze the differentially expressed genes at 24h vs. 48h?
2. Table 2 is too long and not visual, so it is suggested as a subsidiary table with a heat map to show the relevant data.
3. Why did the leaves of the control also turn yellow after the 14-day cold stress treatment in Figure 6?
4. Figures 4 and 6 are blurry and need to be replaced with clearer images.
5. There are some grammatical and detail errors in the article, which are suggested to be revised. For example, line 339 should be CYP79B3 instead of YP79B3.
Minor editing of English language required
Author Response
MANUSCRIPT ID: agronomy-2500918
Title: Comparative Transcriptomics Reveal the Mechanisms Underlying the Glucosinolate Metabolic Response in Leaf Brassica juncea L. Under Cold Stress
Journal: Agronomy
Author: Bing Tang1, Bao-Hui Zhang1, Chuan-Yuan Mo1, Wen-Yuan Fu1, Wei Yang1, Qing-Qing Wang1, Ning Ao1, Fei Qu1, Guo-Fei Tan1, Lian Tao1, Ying Deng1*
Dear Editors and Reviewers,
Thank you and the reviewers very much for revising our manuscript ‘Comparative Transcriptomics Reveal the Mechanisms Underlying the Glucosinolate Metabolic Response in Leaf Brassica juncea L. Under Cold Stress’ (MANUSCRIPT ID: agronomy-2500918). Your effort and time spent on our manuscript are greatly appreciated by all of us. We are delighted to all suggestion and review comments, which you and the reviewers made. Your revisions/suggestions have definitely improved the quality of our manuscript.
This manuscript has been extensively edited according to your and reviewers’ comments. The manuscript has been improved by a professional English language editing service (Author Services-SCINET Co., Ltd; Website: www.scinet.com.cn). Please find the revised manuscript in Agronomy’s manuscript center. The changes were made directly in the text with RED marked. The responses to the reviewers are highlighted below.
Thank you again for your kind help and excellent suggestions for our manuscript. We hope these revisions will be satisfactory. We are looking forward to hearing from you soon.
Yours sincerely
Bing Tang, Guo-Fei Tan, Ying Deng
Institute of Horticulture, Guizhou Academy of Agricultural Sciences/Horticultural Engineering Technology Research Center of Guizhou, Guiyang 550006, China
Reviewer 1:
The manuscript entitled “Comparative Transcriptomics Reveal the Mechanisms Under-lying the Glucosinolate Metabolic Response in Leaf Brassica juncea L. Under Cold Stress” authored by tang et al. is interesting and meaningful. I have read the manuscript and suggest some minor revisions before to accept the paper for publication in Agronomy.
- How come the analysis in the DEG differential analysis in lines 186-201 did not analyze the differentially expressed genes at 24 h vs. 48 h?
Response:
-- We thank the reviewer for this observation and suggestion.
-- We agree. We analyzed the differential expression at 24 h and 48 h at GSLs synthesis-related differentially expressed genes, as follows:
- Table 2 is too long and not visual, so it is suggested as a subsidiary table with a heat map to show the relevant data.
Response:
-- We thank the reviewer for this observation and suggestion.
-- We agree. We appreciate your comments, questions and suggestions that help us to improve our manuscript. We already present the data from Table 2 in Figure 4A as well and put Table 2 inside the supplementary data Table S2.
- Why did the leaves of the control also turn yellow after the 14-day cold stress treatment in Figure 6?
Response:
-- We thank the reviewer for this observation and suggestion.
-- We agree. We appreciate your comments, questions and suggestions that help us to improve our manuscript. The leaves of the control are related to the reduced chlorophyll synthesis under cold stress after 14 d.
- Figures 4 and 6 are blurry and need to be replaced with clearer images.
Response:
-- We thank the reviewer for this observation and suggestion.
-- We agree. We appreciate your comments, questions and suggestions that help us to improve our manuscript. We increased the font size and changed the typography to make the image more visible.
Figure 4. The GSLs synthesis pathway and related genes expression. A. Construction of the GSLs synthesis pathway and synthetic gene expression in B. juncea. B. Evolutionary tree produced using the neighbor-joining method and gene expression of the GSLs degradation-related genes, BGLUs.
Figure 6. Effect of GSLs treatment on cold resistance in leaf B. juncea. A. Phenotypes of GSLs-treated B. juncea and control after 14 d of cold stress; B, C, D, E. Propylene glycol, free proline, soluble sugar and soluble protein contents of GSLs-treated B. juncea and control after 14 d of cold stress; E. Expression of GSLs synthesis-related genes (CYP79B3, CYP79F1, CYP79F2, GSTF9, GSTF10, SUR1, SOT17, SOT18 and MYBS3) in GSLs-treated B. juncea and control after 14 d of cold stress.
- There are some grammatical and detail errors in the article, which are suggested to be revised. For example, line 339 should be CYP79B3 instead of YP79B3.
Response:
-- We thank the reviewer for this observation and suggestion.
-- We agree. We appreciate your comments, questions and suggestions that help us to improve our manuscript, we have been modified. The manuscript has been improved by a professional English language editing service (Author Services-SCINET Co., Ltd; Website: www.scinet.com.cn)
Response: Thank you very much for your comments, they have been modified.
- Comments on the Quality of English Language,Minor editing of English language required.
Response:
-- We thank the reviewer for this observation and suggestion.
-- We agree. We appreciate your comments, questions and suggestions that help us to improve our manuscript. The manuscript has been improved by a professional English language editing service (Author Services-SCINET Co., Ltd; Website: www.scinet.com.cn)

Reviewer 2 Report
The manuscript “Comparative Transcriptomics Reveal the Mechanisms Underlying the Glucosinolate Metabolic Response in Leaf Brassica juncea L. Under Cold Stress” submitted by the Tang et al. was carefully reviewed. Glucosinolates (GSLs) are plant secondary metabolites comprising sulfur and nitrogen mainly found in plants from the order of Brassicales. The degradation products and glucosinolates of GSLs are important substances for plant disease resistance, insect resistance, and stress resistance, which are of great significance for plant growth and development. In their study, transcriptome at different stages were analysised and GSLs content in these samples were tested. Furthermore, the candidate transcription factors related to low-temperature stress were screened and verified. The logical structure of the manuscript needs to be improved and further revisions to be considered for publication.
My main concerns are as follows:
(1) In the text, abbreviations for genes or transcription factors require italics, such as” BGLU25 and BGLU27”, and L245-L280
(2) There is no need to list references in the results section, such as “L247” and “L293”
(3) Table 1 should be moved into supplementary data
(4) In Figure4B, why do the same genes differ significantly in different samples under the same treatment? Especially in the control group (ck)?
(5) Based on transcriptome, why is the gene expression trend between different samples not analyzed, i.e. which genes have the same trend of change under different treatments.
(6) The original data of Transcriptome data need to be uploaded to the public database.
(7) The list of differentially expressed genes among different samples of all transcriptome needs to be put into the supplementary files
(8) In Figure4, All the numbers in the picture are too small to be seen clearly.
Moderate editing of English language required.
Author Response
MANUSCRIPT ID: agronomy-2500918
Title: Comparative Transcriptomics Reveal the Mechanisms Underlying the Glucosinolate Metabolic Response in Leaf Brassica juncea L. Under Cold Stress
Journal: Agronomy
Author: Bing Tang1, Bao-Hui Zhang1, Chuan-Yuan Mo1, Wen-Yuan Fu1, Wei Yang1, Qing-Qing Wang1, Ning Ao1, Fei Qu1, Guo-Fei Tan1, Lian Tao1, Ying Deng1*
Dear Editors and Reviewers,
Thank you and the reviewers very much for revising our manuscript ‘Comparative Transcriptomics Reveal the Mechanisms Underlying the Glucosinolate Metabolic Response in Leaf Brassica juncea L. Under Cold Stress’ (MANUSCRIPT ID: agronomy-2500918). Your effort and time spent on our manuscript are greatly appreciated by all of us. We are delighted to all suggestion and review comments, which you and the reviewers made. Your revisions/suggestions have definitely improved the quality of our manuscript.
This manuscript has been extensively edited according to your and reviewers’ comments. The manuscript has been improved by a professional English language editing service (Author Services-SCINET Co., Ltd; Website: www.scinet.com.cn). Please find the revised manuscript in Agronomy’s manuscript center. The changes were made directly in the text with RED marked. The responses to the reviewers are highlighted below.
Thank you again for your kind help and excellent suggestions for our manuscript. We hope these revisions will be satisfactory. We are looking forward to hearing from you soon.
Yours sincerely
Bing Tang, Guo-Fei Tan, Ying Deng
Institute of Horticulture, Guizhou Academy of Agricultural Sciences/Horticultural Engineering Technology Research Center of Guizhou, Guiyang 550006, China
Reviewer 2:
The manuscript “Comparative Transcriptomics Reveal the Mechanisms Underlying the Glucosinolate Metabolic Response in Leaf Brassica juncea L. Under Cold Stress” submitted by the Tang et al. was carefully reviewed. Glucosinolates (GSLs) are plant secondary metabolites comprising sulfur and nitrogen mainly found in plants from the order of Brassicales. The degradation products and glucosinolates of GSLs are important substances for plant disease resistance, insect resistance, and stress resistance, which are of great significance for plant growth and development. In their study, transcriptome at different stages were analysised and GSLs content in these samples were tested. Furthermore, the candidate transcription factors related to low-temperature stress were screened and verified. The logical structure of the manuscript needs to be improved and further revisions to be considered for publication.
My main concerns are as follows:
(1) In the text, abbreviations for genes or transcription factors require italics, such as” BGLU25 and BGLU27”, and L245-L280
Response:
-- We thank the reviewer for this observation and suggestion.
-- We agree. We appreciate your comments, questions and suggestions that help us to improve our manuscript. We have italicized all abbreviations of genes or transcription factors and labeled them in red.
(2) There is no need to list references in the results section, such as “L247” and “L293”
Response:
-- We thank the reviewer for this observation and suggestion.
-- We agree with your comments. But we need to cite references inside the results to support our next point. For example, the classification of BLGUs genes needs to be explained.
(3) Table 1 should be moved into supplementary data
Response:
-- We thank the reviewer for this observation and suggestion.
-- We agree. We have been modified and turn this form into Table S1.
(4) In Figure4B, why do the same genes differ significantly in different samples under the same treatment? Especially in the control group (ck)?
Response:
-- We thank the reviewer for this observation and suggestion.
-- We agree. We appreciate your comments. We standardized the data (z-scale) in order to more visually express the trend of the expression of the same sample in the different treatments. When some FPKMs are 0, the non-zero data may show images with significant differences, but the data are not significantly different, e.g., BjuVA01G344000: 0.105619, 0, 0, 0, 0.028107, 0, 0.05155, 0.
Figure 4. The GSLs synthesis pathway and related genes expression. A. Construction of the GSLs synthesis pathway and synthetic gene expression in B. juncea. B. Evolutionary tree produced using the neighbor-joining method and gene expression of the GSLs degradation-related genes, BGLUs.
(5) Based on transcriptome, why is the gene expression trend between different
Response:
-- We thank the reviewer for this observation and suggestion.
-- We agree. We add trend analysis to the synthetic analysis of important GSLs at L270-L273 and mark them in red.
(6) The original data of Transcriptome data need to be uploaded to the public database.
Response:
-- We thank the reviewer for this observation and suggestion.
-- We agree. We appreciate your comments, questions and suggestions that help us to improve our manuscript. We have provided the address details in the Data Availability Statement section, specifically NCBI database under BioProject: PRJNA993419.
(7) The list of differentially expressed genes among different samples of all transcriptome needs to be put into the supplementary files
Response:
-- We thank the reviewer for this observation and suggestion.
-- We agree. We appreciate your comments, questions and suggestions that help us to improve our manuscript. The list of differentially expressed genes among different samples of all transcriptome was put into the supplementary files, see Table S3 and Table S4.
(8) In Figure4, All the numbers in the picture are too small to be seen clearly.
Response:
-- We thank the reviewer for this observation and suggestion.
-- We agree. We appreciate your comments, questions and suggestions that help us to improve our manuscript. We have revised the relevant questions one by one based on your comments. We increased the font size and changed the typography to make the image more visible.
Figure 4. The GSLs synthesis pathway and related genes expression. A. Construction of the GSLs synthesis pathway and synthetic gene expression in B. juncea. B. Evolutionary tree produced using the neighbor-joining method and gene expression of the GSLs degradation-related genes, BGLUs.

Reviewer 3 Report
The MS is well-written, but has some shortcomings.
Major concderns:
1. GSLs content increased by 11 % and then decreased by 6%. The results are statistically significantly diffedrent, but do these changes have biological meaning?
2. Figures 4 and 6 are hardly readable
3. The experiment with external application of GSLs is not mentioned in the Materials and Methods at all.
Fig. 6, 332, 437 - the decrease in MDA content is not significant
4. It would be more intersting if you provide the data on MDA, proline, soluble sugars and protein content on Day 0 of the experiment (green plants on Fig.6a). As there is no description of the experiment with exogenous GSLs, it is not clear how plants marked as GSL treatment differ from the control on day 0.
Minor concerns:
30-31: not understandable
32: prefers to be cooled sounds not scientific. Better "optimum growth temperature is ... "
34: cooler than where?
43: Only plant resistancer to insects and pathogens is mentioned, but what about abiotic stress responses? see lines 60-61.
73: was used
79-80: What do you mean by biological replicates if not plants?
Author Response
MANUSCRIPT ID: agronomy-2500918
Title: Comparative Transcriptomics Reveal the Mechanisms Underlying the Glucosinolate Metabolic Response in Leaf Brassica juncea L. Under Cold Stress
Journal: Agronomy
Author: Bing Tang1, Bao-Hui Zhang1, Chuan-Yuan Mo1, Wen-Yuan Fu1, Wei Yang1, Qing-Qing Wang1, Ning Ao1, Fei Qu1, Guo-Fei Tan1, Lian Tao1, Ying Deng1*
Dear Editors and Reviewers,
Thank you and the reviewers very much for revising our manuscript ‘Comparative Transcriptomics Reveal the Mechanisms Underlying the Glucosinolate Metabolic Response in Leaf Brassica juncea L. Under Cold Stress’ (MANUSCRIPT ID: agronomy-2500918). Your effort and time spent on our manuscript are greatly appreciated by all of us. We are delighted to all suggestion and review comments, which you and the reviewers made. Your revisions/suggestions have definitely improved the quality of our manuscript.
This manuscript has been extensively edited according to your and reviewers’ comments. The manuscript has been improved by a professional English language editing service (Author Services-SCINET Co., Ltd; Website: www.scinet.com.cn). Please find the revised manuscript in Agronomy’s manuscript center. The changes were made directly in the text with RED marked. The responses to the reviewers are highlighted below.
Thank you again for your kind help and excellent suggestions for our manuscript. We hope these revisions will be satisfactory. We are looking forward to hearing from you soon.
Yours sincerely
Bing Tang, Guo-Fei Tan, Ying Deng
Institute of Horticulture, Guizhou Academy of Agricultural Sciences/Horticultural Engineering Technology Research Center of Guizhou, Guiyang 550006, China
Reviewer 3:
The MS is well-written but has some shortcomings.
Major concderns:
- GSLs content increased by 11 % and then decreased by 6%. The results are statistically significantly diffedrent, but do these changes have biological meaning?
Response:
-- We thank the reviewer for this observation and suggestion.
-- We agree. These changes are made by biological significance, GSLs are associated with biotic stress, abiotic stress resistance in plants. Changes in the content of GSLs may be one of the countermeasures that can only cope with the stress.
- Figures 4 and 6 are hardly readable
Response:
-- We thank the reviewer for this observation and suggestion.
-- We agree. We increased the font size and changed the typography to make the image more visible.
Figure 4. The GSLs synthesis pathway and related genes expression. A. Construction of the GSLs synthesis pathway and synthetic gene expression in B. juncea. B. Evolutionary tree produced using the neighbor-joining method and gene expression of the GSLs degradation-related genes, BGLUs.
Figure 6. Effect of GSLs treatment on cold resistance in leaf B. juncea. A. Phenotypes of GSLs-treated B. juncea and control after 14 d of cold stress; B, C, D, E. Propylene glycol, free proline, soluble sugar and soluble protein contents of GSLs-treated B. juncea and control after 14 d of cold stress; E. Expression of GSLs synthesis-related genes (CYP79B3, CYP79F1, CYP79F2, GSTF9, GSTF10, SUR1, SOT17, SOT18 and MYBS3) in GSLs-treated B. juncea and control after 14 d of cold stress.
- The experiment with external application of GSLs is not mentioned in the Materials and Methods at all.
Response:
-- We thank the reviewer for this observation and suggestion.
-- We agree. We appreciate your comments, questions and suggestions that help us to improve our manuscript. It has been modified, Added and redlined at L86-93.
- Fig. 6, 332, 437 - the decrease in MDA content is not significant
Response:
-- We thank the reviewer for this observation and suggestion.
-- Thank you very much for your feedback, but our data shows that the that the decrease in MDA content was differentially significant. The raw data is as follows:
CK: 26.12898 15.48384 15.16126 18.06448 22.25802 16.77416
Glucosinolate treatment: 15.48384 14.19352 13.22578 14.5161 13.22578 15.80642 14.5161
- It would be more intersting if you provide the data on MDA, proline, soluble sugars and protein content on Day 0 of the experiment (green plants on Fig.6a). As there is no description of the experiment with exogenous GSLs, it is not clear how plants marked as GSL treatment differ from the control on day 0.
Response:
-- We thank the reviewer for this observation and suggestion.
-- We agree. We appreciate your comments, questions and suggestions that help us to improve our manuscript. We are very sorry, and we are unable to provide data on MDA, proline, soluble sugars and protein content on day 0, because of our small sample size and our concern that mechanical losses caused by leaves collection may affect the expression of genes associated with mustard GSLs. We have supplemented the experimental notes (L86-L93), both of which were treated identically on day 0.
Minor concerns:
30-31: not understandable
Response:
-- We thank the reviewer for this observation and suggestion.
-- We agree. We have changed the description to make it easier to understand, as follows: B. juncea can be categorized on the basis of its edible organs into leaf B. juncea, stem B. juncea, and root B. juncea.
32: prefers to be cooled sounds not scientific. Better "optimum growth temperature is ... "
Response:
-- We thank the reviewer for this observation and suggestion.
-- We agree. We appreciate your comments, it has been modified.
34: cooler than where?
Response:
-- We thank the reviewer for this observation and suggestion.
-- We agree. We appreciate your comments, cooler climate is a fixed collocation phrase meaning a cool climate.
43: Only plant resistancer to insects and pathogens is mentioned, but what about abiotic stress responses? see lines 60-61.
Response:
-- We thank the reviewer for this observation and suggestion.
-- We agree. We appreciate your comments, it has been modified.
Response: Thank you very much for your comment, it has been modified
79-80: What do you mean by biological replicates if not plants?
Response:
-- We thank the reviewer for this observation and suggestion.
-- We agree. We appreciate your comments, the biological replicates is plant, it has been modified.

Round 2
Reviewer 2 Report
In Figure 1, the bar should be added.
Author Response
MANUSCRIPT ID: agronomy-2500918
Title: Comparative Transcriptomics Reveal the Mechanisms Underlying the Glucosinolate Metabolic Response in Leaf Brassica juncea L. Under Cold Stress
Journal: Agronomy
Author: Bing Tang1, Bao-Hui Zhang1, Chuan-Yuan Mo1, Wen-Yuan Fu1, Wei Yang1, Qing-Qing Wang1, Ning Ao1, Fei Qu1, Guo-Fei Tan1, Lian Tao1, Ying Deng1*
Dear Editors and Reviewers,
Thank you and the reviewers very much for revising our manuscript ‘Comparative Transcriptomics Reveal the Mechanisms Underlying the Glucosinolate Metabolic Response in Leaf Brassica juncea L. Under Cold Stress’ (MANUSCRIPT ID: agronomy-2500918). Your effort and time spent on our manuscript are greatly appreciated by all of us. We are delighted to all suggestion and review comments, which you and the reviewers made. Your revisions/suggestions have definitely improved the quality of our manuscript. We appreciate your comments, questions and suggestions that help us to improve our manuscript, we have been added the bar in Figure 1.

Reviewer 3 Report
All my question are answered. Thank you
Author Response
Thank you and the reviewers very much for revising our manuscript ‘Comparative Transcriptomics Reveal the Mechanisms Underlying the Glucosinolate Metabolic Response in Leaf Brassica juncea L. Under Cold Stress’ (MANUSCRIPT ID: agronomy-2500918). Your effort and time spent on our manuscript are greatly appreciated by all of us. We are delighted to all suggestion and review comments, which you and the reviewers made. Your revisions/suggestions have definitely improved the quality of our manuscript.